NxRepair: error correction in de novo sequence assembly using Nextera mate pairs

Murphy Rebecca R. 1
O’Connell Jared 2
Cox Anthony J. 2
Schulz-Trieglaff Ole 2 ole.st@gmx.de
1 Department of Chemistry, University of Cambridge , UK
2 Illumina Cambridge, Chesterford Research Park , Essex , UK
Emes Richard
Electronic publication date: 2015 Jun 2
Publication date: 2015
Volume: 3
Electronic Location ID: e996
Received 2014 Dec 20; Accepted 2015 May 13
Copyright: © 2015 Murphy et al.
Copyright year: 2015
Copyright holder: Murphy et al.
License: This is an open access article distributed under the terms of the Creative Commons Attribution License, which permits unrestricted use, distribution, reproduction and adaptation in any medium and for any purpose provided that it is properly attributed. For attribution, the original author(s), title, publication source (PeerJ) and either DOI or URL of the article must be cited.
License URL: https://creativecommons.org/licenses/by/4.0/

Keywords: De novo assembly, Mate pair, Genome assembly, Error correction, Scaffolding, Insert size, Misassembly, Misassembly detection, Assembly quality, Automated error detection

Funding: RRM is a BBSRC Ph.D. student. This work was completed during a paid internship at Illumina. Jared O’Connell, Anthony J. Cox and Ole Schulz-Trieglaff are permanent employees of Illumina Inc. The funders had no role in study design, data collection and analysis, decision to publish, or preparation of the manuscript.

==============================
Scaffolding errors and incorrect repeat disambiguation during de novo assembly can result in large scale misassemblies in draft genomes. Nextera mate pair sequencing data provide additional information to resolve assembly ambiguities during scaffolding. Here, we introduce NxRepair, an open source toolkit for error correction in de novo assemblies that uses Nextera mate pair libraries to identify and correct large-scale errors. We show that NxRepair can identify and correct large scaffolding errors, without use of a reference sequence, resulting in quantitative improvements in the assembly quality. NxRepair can be downloaded from GitHub or PyPI, the Python Package Index; a tutorial and user documentation are also available.

Introduction

De novo assembly is the construction of a long, contiguous genomic sequence from short DNA reads, without using a reference genome. A common method of de novo genome assembly is construction and traversal of a de Bruijn graph (Compeau, Pevzner & Tesler, 2011) formed by combining overlapping short reads. In the simplest case, the graph is constructed from single end reads. However, with only single end reads, disambiguating repeat regions, which tangle the de Bruijn graph, remains challenging. Paired end read technology (Fullwood et al., 2009), in which a longer fragment of DNA is sequenced from both ends, to create a pair of short reads separated by an unsequenced region. The genomic distance from the start of one read to the end of the other is termed the insert size. Paired end reads with insert sizes of a few hundred bases provide some additional information for repeat disambiguation; even more useful are read pairs with a large insert size of several kilobases, such as the Illumina Nextera mate pairs (Illumina, 2012). Assembly is typically a two-stage process. First, long contiguous sections, named contigs, are constructed. Second, once the contigs cannot be extended any further, scaffolding algorithms attempt to join multiple contigs, using insert size information to determine contig order and approximate gap size. Many assemblers incorporate mate pair insert size information into either both the contig assembly and scaffolding processes (Bankevich et al., 2012), or just into the scaffolding step (Zerbino & Birney, 2008) but errors can still occur. The most serious mistakes are large scale scaffolding or extension errors (Fig. 1A), in which two two disparate regions of a genome are incorrectly joined together. Similarly, large insertion or deletion errors (indels) create structural irregularities in the de novo assembly; whereas mistakes in base calling lead to errors at a single position only.

Figure 1 Using NxRepair to remove large misassemblies.

(A) Alignment of the de novo assembly of the TB genome to its reference genome. The assembly contains several large misassemblies. (B) A plot of NxRepair’s support metric against scaffold position for the TB assembly. Low support for the assembly is identified in three regions of a contig. (C) Breaking the contigs at the identified positions resolves the most significant misassemblies. In (A) and (C), horizontal lines demarcate the scaffold boundaries.

Errors in de novo assembly can significantly affect the quality of the assembled genome, with repercussions for downstream research. Consequently, error correction and quality evaluation of de novo assemblies are problems receiving considerable research interest. Recent work, such as the Assemblathon (Bradnam et al., 2013) and GAGE (Salzberg et al., 2012) collaborations, compare the quality of assemblies prepared by various assemblers by comparing the de novo assemblies produced with their equivalent reference genomes. However, in the absence of a high-quality reference genome, other methods of quality evaluation must be used. Ghodsi et al. (2013) have developed a Bayesian method of assembly quality evaluation, which can calculate an assembly quality score, without requiring a reference genome. However, this provides only an overall quality score and cannot be used to identify errors or low-quality regions. Several recent papers have developed error identification and correction methods, which perform a fine-grained analysis of assembly quality. The most well-known of these is the A5 Assembly Pipeline (Coil, Jospin & Darling, 2014; Tritt et al., 2012), which includes an error detection and rescaffolding step that makes use of mate pair alignment information. Two new tools, REAPR (Hunt et al., 2013) and ALE (Clark et al., 2013) have also been developed to use read pair data to identify misassemblies. A similar tool is currently under development at the Broad Institute (Walker, 2014). However, with the exception of ALE, which is no longer actively maintained, these newer tools are not optimised to use mate pair information.

Here we introduce NxRepair, an assembly error detection tool that can identify the most serious misassemblies by examining the distribution of Nextera mate pair insert sizes. NxRepair does not require a reference genome and can be used with assemblies prepared with just a single mate pair library. It specifically targets the most serious scaffolding errors and large-scale indels by identifying regions with a high number of anomalous insert sizes, or very few supporting reads, breaking the scaffold and optionally trimming out the misassembled region. NxRepair also provides a fine-grained quality score, allowing researchers to visualise poor-quality regions. We demonstrate usage of NxRepair on bacterial genomes assembled from a single Nextera mate pair library using the state of the art SPAdes assembler (Bankevich et al., 2012), which explicitly uses insert size information during contig construction, as well as for scaffolding. Using these genomes, we benchmark NxRepair against the error correction module of the A5 assembler, A5qc (Tritt et al., 2012), which is currently the most widely used error correction tool.

Implementation

Statistical analysis of mate pair insert sizes

Nextera mate pair libraries are prepared to have a certain insert size, typically between 1 and 10 kb. When the mate pairs used to prepare an assembly are aligned back to the assembly, large misassemblies result in unusual insert sizes and read orientations. We model this using a two-component mixture distribution. The first component of this mixture is the insert size distribution of correctly aligned mate pairs. We model the distribution of insert sizes, Y, as a normal distribution with mean μˆ and standard deviation σˆ: Y∼Nμˆ,σˆ2. We estimate μˆ and σˆ for the entire genome by aligning reads back to the assembly and using robust estimators (see below). The second component, defined as a uniform distribution across the contig size U(0, L) for a contig of length L, captures anomalous insert sizes.

To calculate the degree of support for the assembly at each site across a contig, NxRepair retrieves all mate pairs spanning the region [i − W, i + W], of size 2W − 1 at position i on the contig, where spanning is defined to mean that one read ends entirely before the region [i − W, i + W] and the other read begins entirely after (see Fig. 2). The default value of W is 200 bases (see Table 1).

Figure 2 Spanning mate pairs.

Schematic illustrating mate pairs spanning a selected region. Pair 1 (green) spans the indicated region, as both reads align entirely outside of the region i − W, i + W. Pair 2 (cyan) do not span the indicated region, as the left-hand read overlaps the target area.

Table 1 NxRepair parameters.

Parameter	Default value	Meaning	
Imgname	None	Prefix under which to save plots.	
Maxinsert	30,000	Maximum insert size, below which a read pair is included in calculating population statistics.	
Minmapq	40	Minimum MapQ value, above which a read pair is included in calculating population statistics.	
Minsize	10,000	Minimum contig size to analyse.	
Prior	0.01	Prior probability that the insert size is anomalous.	
Stepsize	1,000	Step-size in bases to traverse contigs.	
Trim	4,000	Number of bases to trim from each side of an identified misassembly.	
T	−4.0	Threshold in Z score (number of standard deviations from the mean) below which a misassembly is called.	
Window	200	Window size across which bridging mate pairs are evaluated.	

A uniform distribution was selected to model anomalous insert sizes, as it makes no assumption about the cause of an anomaly. It is uniform over the contig length, L, as opposed to over all possible sites in the assembly, as only pairs where both members align fully to that contig are considered. Similarly, even though the insert size distribution for correctly aligned mate pairs will typically display a longer tail than the normal distribution (unless a gel-extraction protocol is used), we found that using a normal distribution to model correct insert sizes did not adversely affect NxRepair’s error detection. This is because each site is spanned by many mate pairs and the insert size of correctly aligned mate pairs is not correlated with location. Consequently, despite these assumptions, the small fraction of correctly aligned mate pairs with a very large insert size do not lead to false positives in error identification.

We define a latent indicator variable Xl ∈ {0, 1} for each pair of reads, l, which takes the value 1 if the insert size came from the null distribution, and 0 otherwise. Within each window queried, the probability that each retrieved read, rl is drawn from the null distribution is given by: (1) PXl=x|Yl=πxYl|Xl=x∑k=01πkYl|Xl=k

where Yl is the insert size of read pair l, πk is the user defined prior probability of class k and π1 + π0 = 1. The default value of π0 is 0.01 (see Table 1), meaning that in the absence of any insert size information, 99% of read pairs are expected to arise from the null distribution.

Within each window, the total support for a correct assembly at position i can be calculated as: (2) Di=∑l=1NPXl=1|Yl⋅Cl

where the summation is over all read pairs aligning across position i and Cl is an indicator variable, reporting pairing orientation: (3) Cl=1,ifmate pairs have correct orientation and strand alignment0,otherwise.

Within each contig, the contig assembly support mean μD and variance sD are calculated from all reads aligning to the contig, (4) μˆD=∑l=1NDlNsD=∑l=1NDl−μˆ2N.

We use these contig specific mean and variance, rather than the global values, to prevent local variations in coverage from either causing false positives or masking changes in the insert size distribution. Although this reduces sensitivity to misassemblies in very small contigs, it is effective at preventing more damaging false positives. Using these values, the Z-score zl within each queried interval is calculated as: (5) zl=Dl−μˆDsD.

The Z-score is sensitive both to local changes in the insert size distribution, and to large variations in the number of correctly aligned mate pairs, for example caused by a large number of reads with a mate aligning to a different contig. This ensures that NxRepair can identify misassemblies occurring both within and between contigs.

A misassembly is identified if zl < T for a user-defined threshold T (default value −4). This threshold describes the number of standard deviations below the mean assembly support that is required to identify an anomaly. The default value of −4 will flag only positions whose assembly support is less than four standard deviations below the mean level of support.

Global assembly parameters

NxRepair identifies misassemblies by identifying regions where the mate pair insert size distribution differs significantly from the insert size distribution across the entirety of the de novo assembly. Consequently, it is necessary to have a robust estimate of the global mate pair insert size distribution. For calculation of population statistics, mate pairs that align to different contigs are excluded, as are mate pairs with an incorrect strand or pairing orientation and pairs whose insert size exceed a user-defined maximum (maxinsert, whose default value of 30 kb is approximately 10 times the mean insert size for Nextera mate pairs). Pairs whose mapping quality falls below a user specified threshold (minmapq, default value 40) are also excluded, removing reads that are not uniquely mapped from the calculation of global parameters. The global mean μˆ and median absolute deviation (MAD) are calculated across all contigs in the assembly as: (6) μˆ=∑l=1NYlNMAD=medianm|Ym−medianlYl|

where medianl(Yl) is the median insert size of reads with correct pairing behaviour and |Ym − medianl(Yl)| is the absolute value of the residual from the median of the mth on N reads. The standard deviation was then calculated from the MAD, using: (7) σˆ=K⋅MAD

for K = 1.4826.

The MAD is a robust estimator for the standard deviation, as it is not sensitive to outliers, such as the long tail of the mate pair insert size distribution. Using the MAD as an estimator prevents over-estimating the variance of the insert size, allowing anomalously large insert sizes to be correctly identified.

These were then used as the parameters of the null distribution, as described in the main paper.

Interval tree construction

To facilitate rapid lookup of mate pair properties, we construct an interval tree (Cormen et al., 2009) for each contig in the de novo assembly. An interval tree is a data structure that facilitates O(logn + m) lookup of intervals that span a given point or interval, for n total entries and m spanning entries. The interval tree contains the start and end positions of each mate pair aligned to that contig, as well as a flag variable indicating whether that mate pair had correct strand and pairing orientation. Mate pairs where the two reads align to different contigs were excluded. Mapping quality is currently not considered at this stage—reads are retained regardless of mapq score. For each position i for which the Z-score is to be calculated, we perform a stabbing query: the tree is queried with a start position i − W and end position i + W, to retrieve read pairs spanning the interval between positions i − W and i + W (exclusive). The insert sizes of retrieved read pairs are then used to calculate the Z-score for position i. This allows NxRepair to rapidly query positions across a contig to discover the insert size distribution at the queried position. Use of the interval tree significantly increases the efficiency of Z-score calculation, as each pair of reads is fetched only once from the bam file in order to build the tree. All relevant parameters are then stored in the tree for rapid look-up when a position is queried. This has several advantages. Firstly, it is significantly faster than fetching reads only when a position is queried. Secondly, it is more space efficient than a frequency array of all positions on all contigs but does not lose any information about the exact alignment positions. Finally, once construction of the tree is complete, multiple passes across a contig (for example with different spatial resolutions, or using different window sizes) can rapidly be made using the same tree.

Misassembly location and contig breaking

To improve the quality of the de novo assembly, a contig is broken into two separate pieces at the site of a misassembly. The broken ends of the two new contigs can optionally be trimmed by a user defined length (default 4 kb) to remove the misassembled region. Trimming allows removal of the incorrectly assembled regions around a break-point, but can be switched off if a user does not want any sequence to be removed from the assembly. To prevent excessive clipping, misassemblies separated by less than the trimming distance are grouped together, the contig is broken at the start and end of the misassembled region and the misassembled section is discarded. Low-scoring regions within the trimming distance of the ends of contigs are not considered misassemblies, as the high proportion of mate pairs aligning here whose mate maps to a different contig reduces the number of pairs under consideration and hence lowers the observed Z-score. This also ensures that circular molecules, such as small plasmids, which are assembled into a single contig, are not truncated because of mate pairs at either end of the assembly that appear to span the entire contig, but which are spatially close when circularisation is considered.

Availability and dependencies

NxRepair is available for free anonymous download from the Python Package Index (PyPI) here: https://pypi.python.org/pypi/nxrepair. The source code, written in Python, is hosted on GitHub: https://github.com/rebeccaroisin/nxrepair. A full tutorial and API can be found on ReadTheDocs: http://nxrepair.readthedocs.org/en/latest/.

NxRepair makes use of several further open source libraries, specifically:

Numpy (Van der Walt, Colbert & Varoquaux, 2011) (http://www.numpy.org/)

Scipy (Millman & Aivazis, 2011) (http://www.scipy.org/)

Matplotlib (Hunter, 2007) (http://matplotlib.org/)

Pysam (https://pypi.python.org/pypi/pysam), the python wrapper for Samtools

Samtools (Li et al., 2009) (http://samtools.sourceforge.net/)

We installed the numpy, scipy and matplotlib libraries via Anaconda (https://store.continuum.io/cshop/anaconda/).

We have used the Interval Tree implementation from the bx-python library (https://bitbucket.org/james_taylor/bx-python/wiki/Home).

Materials and Methods

Data

Nine bacterial genomes were prepared according to the Nextera mate pair protocol and sequenced in duplicate in two independent MiSeq runs using 2 × 151 bp reads. The organisms sequenced are shown in Table 4. Reads were trimmed using the MiSeq inbuilt trimmer. Table 5 gives an overview of sequencing yield, mean quality and read length after trimming. The untrimmed reads are available from BaseSpace via https://basespace.illumina.com/s/TXv32Ve6wTl9 (free registration required). In addition, the trimmed reads are available at the European Nucleotide Archive (ENA) at http://www.ebi.ac.uk/ena/data/view/PRJEB8559. Note that only these Nextera mate pair libraries were used. No additional single end or paired end libraries were required. For performance optimisation, the first replicate from each genome sequenced was used as a training set. The test set, for performance evaluation, was formed from the second replicate of each genome. The replicates of each genome are derived from the same library, but were from separate sequencing runs, facilitating independent sub-sampling of the fragment library. Hence, there is considerable variation between the two replicates in both the yields and read-lengths, as well as in the de novo assemblies obtained (see Table 5). We are aware that using replicates prepared in this manner does not result in as much variation as if two separate libraries had been prepared for each genome. However, given the considerable resources required to generate such libraries, we feel that this is a reasonable compromise that makes efficient use of the available data.

Performance optimisation

ROC plots

To optimise the threshold in Z below which to identify a misassembled region, we prepared ROC plots using Replicate 1 of each genome, varying the threshold value, T, in steps of 1 between −10 and 0.

The positions of true misassemblies were identified by aligning each de novo assembly to its reference genome using QUAST (Gurevich et al., 2013). To correctly compare the sites of true misassemblies with those identified by NxRepair, we divided each contig of the assembly into short stretches of 1 kb length. We then prepared an array, ANx of size L1000 for contig length L, corresponding to misassemblies identified by NxRepair. ANx was filled as follows: (8) ANx=1,ifNxRepair identified a misassembly in stretch i0,otherwise.

To prepare the ROCs, each position i in ANx was labeled as true positive (TP) if ANx[i] = 1 and a true misassembly fell within it, true negative (TN) if ANx[i] = 0 and no true misassembly occurred within the interval, false positive (FP) if ANx[i] = 1 but no true misassembly had occurred, or false negative (FN) if ANx[i] = 0 but the interval contained a true misassembly. The 1 kb interval used was the same interval used in error identification, ensuring that the resolution of the evaluation matched the error detection resolution. The true positive rate (TPR) and false positive rate (FPR) were then calculated as follows: (9) TPR=TPTP+FNFPR=FPFP+TN.

Based on the resultant ROC plots, shown in Fig. 3, a threshold in Z of −4 was found to detect true misassemblies with minimal false positives, so was used for all subsequent analyses and is the default value used in NxRepair error correction. Users who are concerned about false positives are encouraged to use a more stringent threshold value.

Figure 3 ROC plots for the seven genomes from Replicate 1 that contained misassemblies.

Profiling

Performance analysis was performed on a single core with 8 GB RAM available. Runtime analysis was performed using the Python cProfile module. The memoryprofiler Python module was used to analyse memory usage.

Workflow pipeline

De novo assemblies were prepared using the SPAdes Assembler, version 3.1.1 (Bankevich et al., 2012): spades.py -k 21,33,55,77 -t 4 --hqmp1-12 bacteria.fastq.gz --hqmp1-fr -o assembly

The initial assembly quality was evaluated using QUAST (Gurevich et al., 2013) (version 2.3) to align the de novo assembly to a reference genome: python quast.py -o results_sample -t 16

-R ref/reference.fna sample_new.fasta

Following assembly, the same reads used to generate the assembly were aligned back to the de novo assembly using BWA-MEM (Li, 2013) (BWA version 0.7.10). A sorted BAM file of the resulting alignment was then prepared using SAMtools (version 1.1) (Li et al., 2009): bwa index sample/scaffolds.fasta bwa mem sample/scaffolds.fasta\ -p bacteria.fastq.gz | samtools view -bS - | samtools sort - sample samtools index sample.bam

We identified misassemblies using NxRepair (version 0.13) as follows: python nxrepair.py sample.bam\ sample/scaffolds.fasta sample_scores.csv sample_new.fasta -img_name sample_new

The default parameters used and their meanings are shown in Table 1. These have been optimised for Illumina Nextera mate pair libraries with a mean insert size of approximately 4 kb. For mate pair libraries with a much larger (smaller) insert size, the maxinsert and trim parameters may need to be increased (decreased).

Finally we used QUAST (Gurevich et al., 2013) to evaluate the assembly quality following NxRepair by aligning the de novo assembly to a reference genome as described above.

Results and Discussion

We used NxRepair to correct de novo assemblies from Replicate 2 of each of the nine bacterial genomes described above. Mate pair reads were trimmed, assembled using the SPAdes assembler (version 3.1.1) (Bankevich et al., 2012) and then aligned back to the assembled scaffold using BWA-MEM (Li, 2013). We used QUAST (Gurevich et al., 2013) to evaluate the assembly quality before and after NxRepair correction by aligning to an appropriate reference genome. For all NxRepair analyses, the default parameters, shown in Table 1 were used. Figure 1A shows a misassembled genome that contained several scaffolding errors identified by NxRepair (Fig. 1B). Following NxRepair correction, the most significant structural misassemblies were resolved (Fig. 1C). The improvement following NxRepair correction is shown for all nine genomes in Table 2 (middle column). For two assemblies, errors were removed without reducing NGA50; for one genome, errors were removed but NGA50 was slightly reduced; for six genomes, three of which contained no large errors, no errors were found and the assembly was unchanged. We are not able to correct all misassemblies, as not all misassemblies exhibit a change in Z-score large enough to identify an error against the background score fluctuation caused by the wide insert size distribution of the Nextera mate pairs.

Table 2 Misassembly detection.

Number of large misassemblies and NGA50 as reported by QUAST before and after correction by NxRepair and A5qc.

		Before NxRepair	After NxRepair	After A5qc	
Genome	Genome size	No.	NGA50	No.	NGA50	No.	NGA50	
B. cereus ATCC 10987	5,432,652	0	1,157,846	0	1,157,846	0	1,157,846	
E. coli K-12 substr. DH10B	4,686,137	7	573,003	6	573,003	7	573,003	
E. coli K-12 substr. MG1655	4,641,652	3	693,692	3	693,692	3	693,692	
L. monocytogenes EGDe	2,944,528	0	1,496,613	0	1,496,613	0	1,496,613	
M. ruber DSM 2366	4,839,203	0	2,702,549	0	2,702,549	0	2,702,549	
P. heparinus DSM 2366	5,167,383	1	1,269,147	0	952,558	1	1,269,147	
K. pneumoniae MGH 78578	5,694,894	8	578,813	8	578,813	–	–	
R. sphaeroides 2.4.1	4,602,977	8	2,715,434	8	2,715,434	8	2,715,434	
M. tuberculosis H37Ra	4,411,532	63	186,136	57	186,136	63	186,136	

Table 3 NxRepair performance analysis.

Bacterium	Total time (s)	Memory usage (MiB)	
B. cereus ATCC 10987	78	271	
E. coli K-12 substr. DH10B	123	444	
E. coli K-12 substr. MG1655	70	260	
L. monocytogenes EGDe	97	383	
M. ruber DSM 2366	259	565	
P. heparinus DSM 2366	123	417	
K. pneumoniae MGH 78578	59	227	
R. sphaeroides 2.4.1	190	463	
M. tuberculosis H37RaTB	155	411	

Table 4 Summary of bacteria analysed and the relevant NCBI information on their reference genomes.

There were two repeats of each strain. All 18 samples were prepared with the Nextera mate pair protocol and sequenced in a single MiSeq run using 2 ×151 bp reads. The untrimmed reads we used as input to NxTrim (3.9Gbp in all) are available from BaseSpace via https://basespace.illumina.com/s/TXv32Ve6wTl9 (free registration required). In addition, the trimmed reads are available at the European Nucleotide Archive (ENA) at http://www.ebi.ac.uk/ena/data/view/PRJEB8559.

Abbreviation:	Bcer	
Bacteria:	Bacillus cereus ATCC 10987	
Accession ID:	NC_003909, NC_005707	
NCBI FTP:	ftp.ncbi.nih.gov/genomes/Bacteria/Bacillus_cereus_ATCC_10987_uid57673/	
Abbreviation:	EcDH	
Bacteria:	Escherichia coli str. K-12 substr. DH10B	
Accession ID:	NC_010473	
NCBI FTP:	ftp.ncbi.nih.gov/genomes/Bacteria/Escherichia_coli_K_12_substr__DH10B_uid58979/	
Abbreviation:	EcMG	
Bacteria:	Escherichia coli str. K-12 substr. MG1655	
Accession ID:	NC_000913	
NCBI FTP:	ftp.ncbi.nih.gov/genomes/Bacteria/Escherichia_coli_K_12_substr__MG1655_uid57779/	
Abbreviation:	list	
Bacteria:	Listeria monocytogenes	
Accession ID:	NC_003210	
NCBI FTP:	ftp.ncbi.nih.gov/genomes/Bacteria/Listeria_monocytogenes_EGD_e_uid61583/	
Abbreviation:	meio	
Bacteria:	Meiothermus ruber DSM 1279	
Accession ID:	NC_013946	
NCBI FTP:	ftp.ncbi.nih.gov/genomes/Bacteria/Meiothermus_ruber_DSM_1279_uid46661/	
Abbreviation:	ped	
Bacteria:	Pedobacter heparinus DSM 2366	
Accession ID:	NC_013061	
NCBI FTP:	ftp.ncbi.nih.gov/genomes/Bacteria/Pedobacter_heparinus_DSM_2366_uid59111/	
Abbreviation:	pneu	
Bacteria:	Klebsiella pneumoniae subsp. pneumoniae MGH 78578	
Accession ID:	NC_009648, NC_009649, NC_009650, NC_009651, NC_009652, NC_009653	
NCBI FTP:	ftp.ncbi.nih.gov/genomes/Bacteria/Klebsiella_pneumoniae_MGH_78578_uid57619/	
Abbreviation:	rhod	
Bacteria:	Rhodobacter sphaeroides 2.4.1	
Accession ID:	NC_007488, NC_007489, NC_007490, NC_007493, NC_007494, NC_009007, NC_009008	
NCBI FTP:	ftp.ncbi.nih.gov/genomes/Bacteria/Rhodobacter_sphaeroides_2_4_1_uid57653/	
Abbreviation:	TB	
Bacteria:	Mycobacterium tuberculosis H37Ra	
Accession ID:	NC_009525	
NCBI FTP:	ftp.ncbi.nih.gov/genomes/Bacteria/Mycobacterium_tuberculosis_H37Ra_uid58853/	

Table 5 Sequencing yields.

Yield in bp, mean base quality and average read length after adapter removal, as well as the raw NGA50 score prior to NxRepair analysis for all genomes analysed.

Genome	Yield [bp]	Mean base quality	Mean read length	Raw NGA50	
B. cereus lib1	140,034,231	30.83	120.50	1,157,404	
B. cereus lib2	150,883,336	31.69	124.08	1,157,846	
E. coli DH10B lib1	229,164,175	31.50	127.55	576,143	
E. coli DH10B lib2	167,955,255	31.19	126.49	573,003	
E. coli MG1655 lib1	138,893,204	NaN	104.56	640,732	
E. coli MG1655 lib2	164,490,239	31.93	129.67	693,692	
L. monocytogenes lib1	197,796,210	32.66	129.79	1,496,615	
L. monocytogenes lib2	161,114,700	31.81	125.79	1,496,613	
M. ruber lib1	180,542,545	29.96	123.69	3,095,733	
M. ruber lib2	150,298,958	31.09	129.40	2,702,549	
P. heparinus lib1	186,070,764	32.21	127.30	1,269,259	
P. heparinus lib2	146,448,694	31.32	124.00	1,269,147	
K. pneumoniae lib1	182,614,602	31.86	131.70	577,220	
K. pneumoniae lib2	166,306,322	31.82	130.28	578,813	
R. sphaeroides lib1	184,138,610	30.08	127.99	3,181,390	
R. sphaeroides lib2	210,961,284	30.12	129.79	2,715,434	
M. tuberculosis lib1	211,892,634	30.43	127.37	184,170	
M. tuberculosis lib2	177,615,358	30.06	126.82	186,136	

To benchmark NxRepair’s performance, we also used the A5qc error correction module of the A5 Assembly pipeline (Tritt et al., 2012) (version 20140604) to identify errors in the de novo assemblies: a5_pipeline.pl --begin=4 --end=4 bacteria.fastq.gz bacteria

The results are shown in Table 2 (right hand column). For eight of the nine genomes evaluated, A5qc was unable to detect any errors. For the final genome (K. pneumoniae), A5qc did detect errors, but the contig-breaking process left Quast unable to align the resultant assembly to the reference genome. Re-scaffolding these contigs using the A5 scaffolder: a5_pipeline.pl --begin=4 --end=5 bacteria.fastq.gz bacteria did allow reference alignment, but the assembly contained more misassemblies (15) than the original assembly.

Despite its better performance in this case, it is clear that, like the A5qc module, NxRepair is not able to find all misassemblies present. There are several reasons for this. Firstly, NxRepair’s resolution is limited to relatively large-scale errors, as a very large disruption in mate pair insert sizes over a region of approximately 1 kb is required to significantly reduce the Z-score. Consequently, indel errors with a displacement smaller than 1 kb will not be detected. Secondly, NxRepair is limited by the intrinsic error rate of the mate pair library used. If the insert size distribution has a very wide variance, large fluctuations caused by assembly errors will be masked, making error correction more challenging.

A number of improvements to NxRepair might mitigate these issues. Firstly, NxRepair currently identifies errors using a simple threshold applied to the total assembly support from spanning mate pairs, D. A more rigorous approach would implement a fully probabilistic method of error detection, using the distribution of spanning mate pair insert sizes to evaluate the relative probability of an error.

Furthermore, NxRepair currently uses a user-defined prior probability of incorrect pairing and uses some simplistic thresholding to determine the parameters of the global insert size distribution. It would be possible to implement simultaneous co-estimatation of the mate pair error rate and the insert size distribution. In addition to relieving the user of estimating the error rate of their mate pair library, this would improve the accuracy of parameter estimation for the correct mate pairs, particularly for libraries with a large mate pair error rate. However, this would not necessarily translate into improved accuracy of error detection, as noise from the mate pair library would still mask true errors.

Performance

We evaluated the runtime and peak memory usage of NxRepair on each of the nine genomes analysed. The results are shown in Table 3. The most memory and computationally intensive part of the NxRepair analysis is construction of the interval trees. The size of each interval tree is dependent on the contig size. Consequently, we expect both runtime and memory usage to scale with the size of the largest contig.

Conclusions

NxRepair is a simple error correction module that can be used to rapidly identify and remove large scale errors from de novo assemblies using Nextera mate pair reads. We evaluated NxRepair using de novo assemblies of nine bacterial genomes prepared using the SPAdes assembler, showing that of the six genomes containing misassemblies, three could be improved by NxRepair correction; compared with no improvements made by the A5qc module. SPAdes is the current state of the art in bacterial genome assembly and explicitly uses mate pair information during both contig construction and scaffolding. Even in these excellent assemblies, NxRepair could identify misassemblies and improve the assembly quality. We predict that NxRepair will be even more useful for identifying errors in de novo assemblies where mate pair information was used only at the scaffolding stage. NxRepair is freely available online. It can be downloaded from the Python Package Index (https://pypi.python.org/pypi/nxrepair) and run with a single call from the command line, making it an attractive option for fast evaluations of and improvements to assembly quality. The source code is available on GitHub (https://github.com/rebeccaroisin/nxrepair), facilitating easy incorporation into user assembly pipelines.

We thank Emma Carlson and Niall Gormley (Illumina Cambridge) for preparing the Nextera mate pair libraries.

Additional Information and Declarations

Competing Interests

Author Contributions

Data Deposition

Jared O’Connell, Anthony J. Cox and Ole Schulz-Trieglaff are permanent employees of Illumina Inc., a public company that develops and markets systems for genomic analysis. They receive shares as part of their compensation.

Rebecca R. Murphy, Jared O’Connell and Ole Schulz-Trieglaff conceived and designed the experiments, performed the experiments, analyzed the data, contributed reagents/materials/analysis tools, wrote the paper, prepared figures and/or tables, reviewed drafts of the paper.

Anthony J. Cox conceived and designed the experiments, analyzed the data, contributed reagents/materials/analysis tools, wrote the paper, reviewed drafts of the paper.

The following information was supplied regarding the deposition of related data:

https://basespace.illumina.com/s/TXv32Ve6wTl9;

http://www.ebi.ac.uk/ena/data/view/PRJEB8559.

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
