# Peer review of "NxRepair: error correction in de novo sequence assembly using Nextera mate pairs"

_PeerJ, doi:10.7717/peerj.996_

## Round 0.1 · original submission · Major Revisions

Your article has been reviewed by two experts in the field and they have each suggested changes that are required for acceptance.

·

Basic reporting

The manuscript describes an algorithm and corresponding software implementation to detect misassemblies in de novo genome assemblies using mate pair sequence data. de novo genome assembly is a highly active area of research, driven by ongoing advances in sequencing technologies. Many of the current generation of assemblers are prone to misassembling regions of genomes that contain high identity repetitive elements, especially those that are at or above the read length or in some cases the size of k-mers used for de bruijn graph-based algorithms. It is exciting to see new efforts to solve these problems.
The context in which a tool such as this would be used could be better introduced. One case is when the initial draft assembly algorithm is unable to incorporate mate-pair information, and a subsequent scaffolding step is to be carried out. In this situation, if the initial assembly contains errors, the scaffolder will be unable to accurately scaffold the assembly with mate-pair data unless the errors are detected and corrected prior to scaffolding. This approach is used for example in the A5 and A5-miseq pipelines. However, it is possible in principle to construct an assembler which leverages the mate pair information directly during the contigging process to avoid such errors in the first place. For such assemblers, a tool like this may not provide any added utility. This kind of information on the scope of applicability could be better introduced.

Experimental design

Overall I think enough of the algorithm was described to understand how it works, however I have a number of questions about the rationale for the design of the method which I have detailed below in the General Comments section. Unfortunately I'm afraid that the design of the accuracy evaluation experiment leaves us with little idea of the method's expected behavior on real datasets. This is because the method appears to have been tuned (T parameter, ROC curves) on the same data for which accuracy is reported. If this is not the case, then the manuscript text needs to be revised to clarify the issue. I am also concerned that no effort has been made to compare the method's performance to previous work to solve the same problem, for example the A5qc module used by the A5 pipeline to detect and correct misassemblies. It is erroneously stated in the introduction that no other software is optimized to work with mate-pair data, yet A5qc does and in fact the use of mate-pairs to detect misassembled contigs was the main motivating use case in its development. The A5 manuscript (Tritt et al 2012 PLoS ONE) discusses its use with mate pair datasets explicitly. A5qc is by no means perfect and is likely to be a bit challenging to use independently of the rest of the pipeline, but that is no excuse for inaccurately representing its application scope and neglecting to benchmark it. Indeed, I suspect A5qc may be able to detect some misassemblies that the present method can not, because the present method is limited to identifying misassemblies where read pairs map entirely within a single contig, whereas A5qc can identify misassemblies involving read pairs that map to different contigs. I would also guess that the present method may be more sensitive than A5qc for the within-contig misassemblies. There could be other tools that are relevant for comparative benchmarking; I have not kept up with the literature in this area.

Validity of the findings

The main issue potentially impacting the validity of the findings is whether the test data were also used for selection of the T parameter. The test datasets are relatively limited, comprising less than 10 genomes, which leaves a non-negligible potential for parameter overfitting.

Note that I did not (yet) evaluate the software itself by running it on my own datasets, in light of the other issues that I think should be addressed first.

Additional comments

The following are specific notes that I made while reading the manuscript:

Abstract: de bruijn assemblers are clearly the most prevalent for Illumina data, but is the scope of applicability really limited to de bruijn assemblers?

Introduction, paragraph 2: The Tritt et al 2012 citation is much more appropriate for A5 in this context, as it describes a technique to use mate pair reads to detect and correct assembly errors. The Coil et al 2014 paper does not include details of that technique, nor does it contain any revisions of that technique.

"Statistical analysis of mate pair insert sizes" paragraph 1: The use of a uniform to model the mate-pair noise makes sense, but it's not clear how appropriate the normal would be for nextera, unless a tight fragment size range has been gel extracted. If data from gel extraction is expected it should be mentioned, and some further discussion is warranted in either case.

same section, paragraph 2: the definition of spanning is somewhat ambiguous. do the read pairs spanning site i include only read pairs with one read entirely before i and the other entirely after? or are pairs where one or both reads actually include i considered 'spanning'? and for a region to be spanned, which of these definitions is correct?

same section, eqn 1: an extra set of P's in the second terms of the numerator & denom would help improve clarity, also there's no need for the first equation -- why not just define the notation for the prior \pi_x up-front and give eqn 2?

same section, eqn 2: The same insert size on a contig of different lengths would give different values for this expression, because the uniform distribution has been defined only over the contig length. Is that intentional? Desirable? The rationale for this decision is not obvious and some explanation is in order.

same section, paragraph 3: It seems like the same process used to estimate the insert size distribution parameters would naturally also yield an estimate of \pi_0. Is that used? I have seen mate pair error levels as high as 20% in some libraries. It's highly sensitive to the lab conditions, so would be ideal to estimate from the data rather than use a fixed value.

same section, eqn 4: what about circular molecules, e.g. plasmids? a large number of pairs near the contig ends will appear to be in the wrong orientation...

same section, eqn 5: why do we need to calculate these contig-specific distributions? is it to deal with local deviation in coverage? or is it because the distributions used in eqn. 1 have a contig-local domain? or something else? some explanation is needed.

same section, eqn 6: it's a bit strange that this approach starts out with the Bayesian framework (e.g. the use of a prior probability) then goes into a frequentist framework for the hypothesis test. One way to keep it Bayesian would be to set up a Bayes factor of the competing hypotheses of no misassembly vs. one or more misassemblies in the target window. then the threshold T could be applied to the Bayes factors instead of the standard deviation.

section "Global Assembly Parameters", eqn 7: the notation for MAD is not quite right, as it suggests taking the median of a single data point in Y.

same section, eqn 7: at high enough levels of noise in the mate pair library, this approach is likely to overestimate the true insert size to some extent. The 30kbp threshold will help mitigate the problem, and the later steps to identify extreme divergence from a contig's background will also reduce the noise, but this likely comes with a cost in sensitivity for misassembly detection.

same section eqn 8: why use this approach instead of one of the other common approaches to calculate standard deviation?

section "Interval Sequence Tree construction": How is the interval sequence tree used in this algorithm? it's not clear at what step in the breakpoint detection the IST gets queried. This should either be explained or the whole section omitted.

section "ROC plots" eqn 10: The FPR here is sensitive to how finely the contigs are sliced into windows because it includes true negatives.

section "Workflow pipeline": nice to see the precision here: exact version & command line

same section: what version of QUAST? they do change a little from one release to the next.

same section, bwa commands: again, versions would be good, even better would be a script that reproduces all the results!

same section nxrepair.py commands: you might want to indicate which version of NxRepair produced the results described here in case you ever fix bugs or make improvements...

·

Basic reporting

- The introduction refers to "assembly errors" but does not distinguish between types of errors, like SNPs, indels, or contig joining mistakes
- No explanation of "insert size" or "mate pair" and "paired end" is given, many readers may not understand these concepts
- No reference for "Nextera Mate Pair" is given
- Existing tools REAPR and ALE are described, and a "Bayesian" method is mentioned but no motivation provided for its mentioning
- The phrase "de novo" should be italicised
- "de Bruijn Graph" should be lowercase "graph"
- missing space at "W is 200 bases"
- interval [i-W,i+W] is 2W+1 not 2W as reported

Experimental design

- It is not clear if you re-sequenced the exact same strains as the reference genomes in NCBI and where these strains were obtained from.
- Versions of software (bwa, samtools, etc) need to be reported
- BWA was used with default parameters, which includes lots of partially mapped reads and alternative mappings. It is unclear how nxRepair handled these.
- It should be made clearer that you are using the same reads for both assembly and post-assembly correcting

Validity of the findings

- The sequencing data is only available on Illumina BaseSpace. This needs to be rectified by placing the reads into a Study on NCBI SRA or into ENA so they are guaranteed to be publicly available.
- Table 1 can be improved by adding in the full species name, the genome size, and the global mate pair statistics that were estimated
- Some measure of the yield, quality and average read length (after clipping) should be provided
- It is claimed the nxRepair fixed 6 of 9 genomes, but Table 1 shows only changes to 3 of the 9 genomes?

Additional comments

- Could this method be incorporated into Spades? Spades already re-aligns the reads back with BWA to correct some errors, so adding in a MP consistency check would be good.
- Do you really need the interval tree data structure, or could the stats you need be computed in a 1-pass manner?
- The use of a uniform distribution for the non-MP reads was interesting. I would have thought most non-MP reads were shadow PE reads, so their distribution would be Gaussian with a low mean and smaller standard deviation, rather than uniform.
- When you break an identified mis-assembly, the trimming part concerns me. Does this mean you are removing a chunk of genomic DNA from the final result? So we could lose genes?

---

## Round 0.2 · Minor Revisions

The reviewer has given a very thorough review and has suggested minor changes. The major concern is the evaluation of the method and the clarity between training and test data sets.

I agree that the distinction is important and will change the likely belief in the method. I hope that you are able to make these changes and look forward to an updated manuscript. If you fell that you are unable to make these changes please contact me or the editorial office to discuss.

·

Basic reporting

The revised version of this manuscript has gone a very long way to resolving my previous concerns with the quality of the exposition and basic reporting. Nearly all of my comments were addressed, and not just with token replies, but with substantive manuscript changes.

Experimental design

I can see that an effort has been made to separate training and test datasets. It looks like this has been done retrospectively, as all datasets in the revised manuscript came from the same sequencing run. It would help to clarify whether the replicate libraries represented separate replicate tagmentation reactions, or the same tagmentation products that were assigned two different barcodes in sequencing. Knowing this will help readers understand which aspects of the libraries can be expected to vary. For example, if two different tagmentation reactions were done per genome, we can expect more variation in insert size distribution between the replicates than if a single tagmentation reaction was carried out.

That said, I'm concerned that these efforts still do not really address the issue of training set and testing set separation. The replicate libraries are likely to have extremely similar characteristics (at least based on my own experience with nextera data they would). And so effectively the method is still trained on the test set. I appreciate that generating these datasets is very time consuming and expensive, and the manuscript already includes more datasets than many other publications in the field analyze. Nevertheless, I find it difficult to accept the accuracy measures as valid given the parameter learning approach employed. One way to get around the issue might be to use some kind of N-fold cross validation scheme, where the data gets repeatedly broken into different subsets for training & testing. Another possibility might be to simply state that sufficient test data are not available to adequately characterize the method's performance, though this would be unfortunate.

The results with A5qc are interesting and not entirely surprising (to me). That NxRepair is potentially able to improve on A5qc's misassembly detection is a great result, especially if it holds up once a clean separation between test and training data has been achieved. The revised manuscript also does a nice job of describing the limitations of using nextera mate pair data for misassembly detection and gives suggestions for how the power to detect misassemblies might be improved in future work.

Validity of the findings

See above in experimental design for remaining concerns regarding validity.

Additional comments

#### Software testing
I was able to install the software with a single command on my ubuntu 14.04LTS laptop:
`sudo pip install nxrepair`
though I had previously installed all the dependencies to use with other software so my experience might not reflect fully what others experience.

I then ran it without any arguments and got the following usage info:

usage: nxrepair [-h] [-min_size min_size] [-img_name img_name] [-trim trim]
[-T T] [-step_size step_size] [-window window]
[-minmapq minmapq] [-maxinsert maxinsert] [-fraction fraction]
[-prior prior]
bam fasta outfile newfasta
nxrepair: error: too few arguments


Running with `--help` provides more detailed help info. So far so good.

I then obtained some data from the [Lynch et al 2014 PLoS Genetics](http://journals.plos.org/plosgenetics/article?id=10.1371/journal.pgen.1004784) study on halophilic archaea genomes, where we generated 6Kbp (gel extracted) mate pair data for a large number of archaeal species. I tested isolate JCM8877. nxrepair appears to run, repeatedly producing a bunch of increasing numbers on the console output, then lists out the scaffolds and reports that it couldn't find any Z-score anomalies, eg. no misassemblies. This was a final assembly produced by A5 (not A5-miseq), so perhaps that's to be expected. To check this further I then tested on the crude scaffolds, corresponding to step 3 in A5, which are more likely to contain misassemblies because the original IDBA was error prone. On that data NxRepair does appear to find two misassemblies, and breaks the scaffolds at these locations, but apparently the resulting scaffolds were below some length limit and so they were removed from the assembly entirely. In terms of user experience, the implementation seems reasonable.


#### Other minor issues
I found the following minor issues while reading the manuscript:

+ Page 1 "De Bruijn" -> "de Bruijn"

+ "We use these contig-specific mean and variance, rather than the global values..." How does this approach behave in small contigs? It seems like the estimates of mean and variance would be subject to much greater variance but I don't have a sense of whether that might be troublesome.

+ "...whose insert size exceed 30 kb..." this is a reasonable default for nextera mate pairs but some companies, e.g. Lucigen offer 40Kbp mate pair kits. Happily this parameter looks to be accessible from the command line, although I did not test it.

+ "...lookup of intervals that span a given point or interval..." I think this is often referred to as a stabbing query

+ "hlThe 1 kb interval used..." typo

+ "...we also used the A5qc error correction module..." It would be helpful if version numbers and command lines could be provided here, as they were in the previous sections, for the sake of reproducibility. It would also help distinguish these results from any future A5qc revision which may behave differently to what was benchmarked here.

---

## Round 0.3 · accepted · Accept

Thank you for clarrifying the limitations on sampling, the concern during the review.